# Wellbeing and coping of UK nurses, midwives and allied health professionals during COVID-19-a cross-sectional study

Patricia Gillen[1,2][�]*, Ruth D. Neill[ID][3☺], John Mallett[ID][4‡], John Moriarty[5‡], Jill Manthorpe[6‡], Heike Schroder[7‡], Denise Currie[7‡], Susan McGrory[1‡], Patricia Nicholl[5‡], Jermaine Ravalier[8‡], Paula McFadden[3]

1 School of Nursing Jordanstown Campus, Ulster University, Newtownabbey, Northern Ireland, United Kingdom, 2 Southern Health and Social Care Trust, Portadown, Northern Ireland, United Kingdom, 3 School of Applied Social and Policy Sciences, Ulster University, Derry-Londonderry, Northern Ireland, United Kingdom, 4 School of Psychology, Ulster University, Coleraine, Northern Ireland, United Kingdom, 5 School of Social Sciences, Education and Social Work, Queen's University Belfast, Belfast, Northern Ireland, United Kingdom, 6 NIHR Policy Research Unit in Health and Social Care Workforce, King's College London, London, United Kingdom, 7 Queen's Management School, Queen's University Belfast, Belfast, Northern Ireland, United Kingdom, 8 School of Science, Bath Spa University, Bath, United Kingdom

☺ These authors contributed equally to this work.
‡ JM, JM, JM, HS, DC, SM, PN, and JR also contributed equally to this work.
* patricia.gillen@southerntrust.hscni.net

**Data Availability Statement:** All relevant data for this study is is included in the paper and here in Ulster University Repository: Gillen, P., Neill, R., Mallett, J., Moriarty, J., Manthorpe, J., Schroeder,

## Abstract

Nurse, Midwives and Allied Health Professionals (AHPs), along with other health and social care colleagues are the backbone of healthcare services. They have played a key role in responding to the increased demands on healthcare during the COVID-19 pandemic. This paper compares cross-sectional data on quality of working life, wellbeing, coping and burn-out of nurses, midwives and AHPs in the United Kingdom (UK) at two time points during the COVID-19 pandemic. An anonymous online repeated cross-sectional survey was conducted at two timepoints, Phase 1 (7th May 2020-3rd July 2020); Phase 2 (17th November 2020-1st February 2021). The survey consisted of the Short Warwick-Edinburgh Mental Wellbeing Scale, the Work-Related Quality of Life Scale, and the Copenhagen Burnout Inventory (Phase 2 only) to measure wellbeing, quality of working life and burnout. The Brief COPE scale and Strategies for Coping with Work and Family Stressors scale assessed coping strategies. Descriptive statistics and multiple linear regressions examined the effects of coping strategies and demographic and work-related variables on wellbeing and quality of working life. A total of 1839 nurses, midwives and AHPs responded to the first or second survey, with a final sample of 1410 respondents -586 from Phase 1; 824 from Phase 2, (422 nurses, 192 midwives and 796 AHPs). Wellbeing and quality of working life scores were significantly lower in the Phase 2 sample compared to respondents in Phase 1 (p<0.001). The COVID-19 pandemic had a significant effect on psychological wellbeing and quality of working life which decreased while the use of negative coping and burnout of these healthcare professionals increased. Health services are now trying to respond to the needs of patients with COVID-19 variants while rebuilding services and tackling the backlog of normal care

H., Currie, D., McGrory, S., Nicholl, P., Ravalier, J., McFadden, P. 2022. Well-being and Coping of UK Nurses, Midwives and AHPs- a cross-sectional study. Ulster University. https://doi.org/10.21251/3f3156a9-8cd4-45e6-b490-13ee74093281.

**Funding:** This work was supported by Health and Social Care Research and Development Division of the Public Health Agency, Northern Ireland (COVID Rapid Response Funding Scheme COM/5603/20), the Northern Ireland Social Care Council (NISCC) and the Southern Health and Social Care Trust, with support from England's National Institute for Health and Care Research (NIHR) Policy Research Unit in Health and Social Care Workforce - PR-PRU-1217-21002.

**Competing interests:** The authors declare no conflict of interest. The funders had no role in the design of the study; in the collection, analyses, or interpretation of data; in the writing of the manuscript, or in the decision to publish the results.

provision. This workforce would benefit from additional support/services to prevent further deterioration in mental health and wellbeing and optimise workforce retention.

## Introduction

Nurse, Midwives and Allied Health Professionals (AHPs), along with other colleagues from health and social care, are at the forefront of healthcare provision, meeting the essential needs of the public in hospital, community and domiciliary settings. The pressures on the UK National Health Service (NHS) are well known, with demand exceeding available funding [1], staff shortages [2], retention and recruitment challenges [3] and increasing waiting lists and times for patients [2]. From early 2020, these pressures were further exacerbated by the COVID-19 pandemic which caused disruption to normal provision and increased pressure, stress and workload particularly for frontline staff.

Healthcare staff have reported concerns about working during COVID-19 including the risk of taking infection home to their family, the need for appropriate personal protective equipment (PPE) and relevant organisational support [4–7]. In addition, there is evidence that infectious disease outbreaks increase stress for healthcare workers [8, 9] with the traumatic impact of working in healthcare during COVID-19 being likened to being exposed to an exceptional trauma; outside a normal human experience, particularly in its exposure to death and dying [10]. In the UK, 850 healthcare workers died of COVID-19 (from March and December 2020) and more than 3000 deaths have been recorded in the United States (US) [11]. The impact of working during COVID-19 was recently reflected in adverse mental health and wellbeing of healthcare workers in three designated regions in China where symptoms of depression (50%), anxiety (45%), insomnia (34%), and distress (72%) were reported by nurses, physicians and other healthcare workers [12]. A meta-analysis of studies by Batra et al., [13] which explored the impact of the COVID-19 pandemic on health care workers' psychological wellbeing reported prevalence rates of anxiety (34.4%), depression (31.8%), stress (40.3%) and insomnia (27.8%).

Healthcare work has been described as emotional labour [14, 15], with the restrictions imposed during the pandemic on visiting to health care facilities and patients' own homes increasing this emotional burden further [16] and impacting staff's ability to cope with increased work demands [15]. Visiting restrictions have led post-operative patients to report less satisfaction with their experience and with their hospital stay overall [17]. Caring for patients who were dying with their loved ones absent or only being virtually present often required nurses, AHPs and other healthcare staff to act as facilitators to those end-of-life inter-actions [18]. This has led to vicarious traumatization not only for the general public but also healthcare staff, including those who may have been less well prepared or trained for the challenges that the pandemic would bring [19].

There is little evidence about how COVID –19 has impacted on midwives. UK midwives provide care for all women during pregnancy, labour, birth and postnatally. Prior to the pandemic, UK midwives (n = 1997) reported experiencing high levels of stress, burnout, anxiety and depression [20]. However, from Australia, Bradfield et al. [4] detailed changes in maternity care practice in response to the pandemic, including reductions in unnecessary admissions or interventions such as induction of labour and an increase in home births and women choosing to freebirth (without a midwife or doctor in attendance), in order to avoid going into hospital. The New Zealand College of Midwives also reported that midwives' working lives

had changed as a result of increased working hours, responding to increased reassurance for women and their families and the need to wear Personal Protective Equipment (PPE) leading to care provision taking longer than usual [21]. In the UK, discussions around restrictions placed on birth partners accompanying women to appointments, during labour and post birth, often at the discretion of individual Trusts [22], provided some insight into the additional stresses placed not only on women and families but also midwives. Bradfield et al. [4] also report the concerns that midwives had in relation to problems being missed due to reduced contact with mothers and their babies. However, not all changes were negative. These midwives also reported benefits of restricted visiting such as more rest for mothers and babies, and midwives being able to spend more time with them while in hospital. In addition, changes in practice including the use of technology to interact with women, for example provision of breastfeeding support, were also considered beneficial and should be sustained post- pandemic.

Coto et al. [23] examined the interrelationship for AHPs between work environment, access to PPE, and levels of stress during COVID-19 in the US. Service delivery models were revised with often hybrid models of in person and tele-health care evolving in response to pandemic related restrictions and patients' needs. As expected, risks of acquisition and transmission of COVID-19 both at work and home caused concern with lower levels of stress reported by those who had access to appropriate PPE than those who did not. Support of psychological health and wellbeing was deemed important and thought to have ameliorated stress in those with access to mental health support services.

Some UK healthcare staff members were required to redeploy during COVID-19 with advice from workplace unions such as the Royal College of Nursing (RCN) providing guidance for members if they had concerns about their competence to work in another area or specialism [24]. Sykes and Pandit [25] reported increased levels of stress and anxiety among redeployed doctors, and Shannon et al. [26] reported increased levels of worry and stress among redeployed health and social care staff in Northern Ireland. Also, staff concerns about redeployment included uncertainty about the role, whether they had the necessary skills, increased risk of personal and family exposure to COVID-19 and increased workload. In February 2021, 29% reported the experience of redeployment as stressful/very stressful, a reduction from first data collection in November 2020 at 38% [26].

Nurses, Midwives and AHPs expertise and skills are central to healthcare provision and given the additional burden on health services because of COVID-19, it is important to examine their quality of working life, mental wellbeing, and coping strategies they used. This may help employers and the workforce to better understand what lessons can be learned and how best to support staff as health services are rebuilt to meet the needs of patients with new, suppressed or diverted problems.

## Study aim

This paper reports the findings of a study to examine the mental wellbeing, coping strategies, burnout and quality of working life of nurses, midwives and AHPs working throughout the UK at two separate time points during the COVID-19 pandemic (May–July 2020) and (November 2020–February 2021). Other papers have reported overall results [27] and findings specifically related to social care workers [28].

## Methods

### Study design and participants

Data from this study are part of a larger ongoing research project entitled 'Health and Social Care Workers' Quality of Working Life and Coping while Working During the COVID-19

Pandemic' launched in May 2020. The research aims to explore the impact of the COVID-19 pandemic on the health and social care workforce, i.e. nurses, midwives, allied health professionals (AHPs), social care workers and social workers in the UK working in a range of settings such as hospitals, care homes (including nursing homes), community/domiciliary and day services. This wider study utilized a repeated cross-sectional design, and the data presented in this current paper were collected at two different time periods; May-July 2020 (Phase 1 of study) and November-February 2021 (Phase 2 of study). The survey was available across Northern Ireland, England, Scotland and Wales. Respondents were recruited by convenience sampling through emails, newsletters and social media posts of employers, regulatory bodies, professional communications, professional associations and workplace unions. Participation in the study at each time point was voluntary and the data were collected through an online survey hosted on the Qualtrics platform.

Study eligibility was based on participants self-reporting their occupation. There was a total of 3290 responses in Phase 1 (7[th] May 2020-3[rd] July 2020) and 3499 responses in Phase 2 (17[th] November 2020-1[st] February 2021) overall, with responses from 1839 graduate entry and regulated healthcare professionals, namely AHPs (387 & 638), Nurses (198 & 361) and Midwives (180 & 75) in Phases 1 and 2 of the study respectively (Fig 1). Demographic and work-related characteristics of the final sample of nurses, midwives and AHPs (n = 1410, Phase 1: 586, Phase 2: 824) included in the present study for Phases 1 and 2 are presented in Table 1.

## Ethical considerations

Ethical approval was obtained from the Research Ethics Filter Committee of the School of Nursing at Ulster university (Ref No: 2020/5/3.1, 23 April 2020, Ulster University, IRAS, Ref No. 20/0073) (for both phases of the study and Trust Governance approval was gained from Health and Social Care Trusts for Phase 2. This allowed the link to the questionnaire in Phase 2 to be shared with HSC Staff via Trust emails. Permission for the use of the scales used in the questionnaire was provided by original authors, and consent and confidentiality were addressed in participant information materials.

## Measures

**Demographics and work-related characteristics.** The anonymous online survey asked respondents about their demographic and work-related characteristics. These variables were consistently measured across Phase 1 and Phase 2 of the wider study; relevant to the current analyses are sex, age, ethnicity, country of work, occupational group, redeployment, disability, and years of experience.

**Mental wellbeing.** Mental wellbeing was assessed using the short version of the Short Warwick-Edinburgh Mental Wellbeing Scale (SWEMWBS) [29], a positively worded seven-item scale assessing statements about thoughts and feeling which asks respondents to describe their experiences of how often they felt this way in the last two weeks. Examples of the scale items are: I've been dealing with problems well, I've been feeling relaxed. A five-point Likert scale ranging from 1 = *None of the time* to 5 = *All of the time* to measure how often is used to rate the items. The scores are summed and transformed into metric scores conversion table [29]. Total scores range from 7 to 35, with higher scores indicating better wellbeing. The scale has good psychometric properties [30, 31] and, in the current study, internal consistency was acceptable (Phase 1: α = 0.86, Phase 2 α = 0.87).

**Work-related quality of life.** Quality of working life was assessed with the 24-item Work-Related Quality of Life Scale (WRQoL) [32]. A five-point Likert scale ranging from 1 = *Strongly disagree* to 5 = *Strongly agree* was used by respondents to indicate their attitudes to the factors

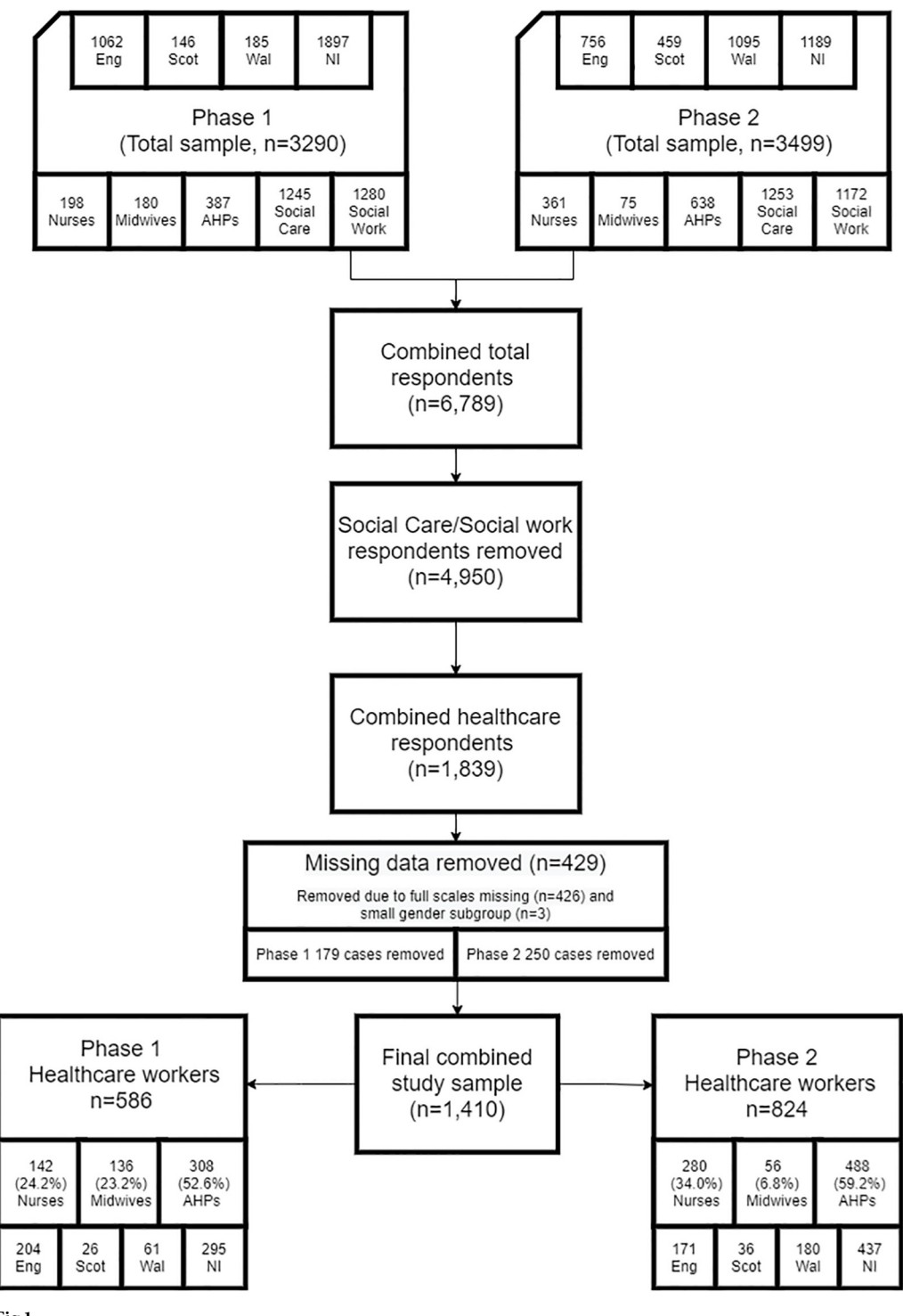

**Fig 1.**

that influenced their quality of working life. Twenty-three items contribute to the overall WRQOL score (with item 24 'overall' excluded from total score) and three items were reverse-scored. In addition to the overall quality of working life, the scale assesses six domains of quality of working life; Job career satisfaction (being content with one's job and career prospects),

**Table 1. Demographics and work-related characteristics of nurses, midwives and AHPs respondents.**

| Variable | Phase 1 (7th May– 3rd July 2020) N = 586 | Phase 2 (17th November 2020 – 1st February 2021) N = 824 |
|---|---|---|
| *Sex* | | |
| Female | 533 (91.0%) | 748 (90.8%) |
| Male | 53 (9.0%) | 76 (9.2%) |
| *Age* | | |
| 16–29 | 74 (12.6%) | 93 (11.3%) |
| 30–39 | 117 (20.0%) | 197 (23.9%) |
| 40–49 | 183 (31.2%) | 222 (26.9%) |
| 50–59 | 177 (30.2%) | 241 (29.2%) |
| 60–65 | 34 (5.8%) | 60 (7.3%) |
| 66+ | 1 (0.2%) | 11 (1.3%) |
| *Ethnic background* | | |
| White | 561 (96.1%) | 797 (97.0%) |
| Black | 7 (1.2%) | 8 (1.0%) |
| Asian | 8 (1.4%) | 4 (0.5%) |
| Mixed | 8 (1.4%) | 13 (1.6%) |
| *Country of work* | | |
| England | 204 (34.8%) | 171 (20.8%) |
| Scotland | 26 (4.4%) | 36 (4.4%) |
| Wales | 61 (10.4%) | 180 (21.8%) |
| Northern Ireland | 295 (50.3%) | 437 (53.0%) |
| *Occupational group* | | |
| Nursing | 142 (24.2%) | 280 (34.0%) |
| Midwifery | 136 (23.2%) | 56 (6.8%) |
| Allied Health Professionals | 308 (52.6%) | 488 (59.2%) |
| *Number of years of work experience* | | |
| Less than 2 years | 35 (6.0) | 45 (5.5%) |
| 2–5 years | 76 (13.0%) | 88 (10.7%) |
| 6–10 years | 86 (14.7%) | 120 (14.6%) |
| 11–20 years | 152 (25.9%) | 230 (27.9%) |
| 21–30 years | 127 (21.7%) | 168 (20.4%) |
| More than 30 years | 110 (18.8%) | 173 (21.0%) |
| *Disability status* | | |
| Yes | 48 (8.2%) | 63 (7.6%) |
| No | 532 (90.8%) | 746 (90.5%) |
| Unsure | 6 (1.0%) | 15 (1.8%) |
| *Redeployment* | | |
| Yes | 118 (20.1%) | 180 (21.8%) |
| No | 468 (79.9%) | 644 (78.2%) |

Stress at work (seeing work pressures as acceptable or excessive), Working conditions (being satisfied with one's working conditions), Control at work (being involved in decisions that affect one's work), General wellbeing (general psychological and physical health) and Home-work interface (whether the organisation helps one with pressures outside of work). Higher overall scores as well as higher scores on the individual domains indicate better quality of working life. The scale has demonstrated good psychometric properties [32, 33] and in the current study internal consistency of the 23 items was good (Phase 1: α = 0.88, Phase 2 α = 0.89).

**Coping.** Coping was assessed using items from two different scales to examine coping with COVID-19-related occupational demands and coping with work-related stressors. A selection of 20 items from the Brief COPE scale [34] assessed ten different coping strategies (active coping, planning, positive reframing, acceptance, use of emotional support, use of instrumental support, venting, substance use, behavioural disengagement, self-blame). Respondents were asked to indicate how often they have been using the strategies described in the items using a four-point Likert scale ranging from 1 = '*I haven't been doing this at all*' to 4 = '*I've been doing this a lot*'. Scores for each coping strategy can range from 2 to 8 and higher scores indicate that respondents use the specific coping strategy more often. Each coping strategy is assessed with two items, which were summed to give a total score. Cronbach's alpha for the 20 items scale was acceptable in the current study (Phase 1: $\alpha = 0.82$, Phase 2 $\alpha = 0.82$), the subscales were rated between 0.69 and 0.91 for reliability between both phases.

The 15 items from the Strategies for Coping with Work and Family Stressors Scale designed by Clark, Michel, Early and Baltes [35] were used to assess five different coping strategies (family-work segmentation, work-family segmentation, working to improve skills/efficiency, recreation and relaxation, exercise). Respondents used a six-point Likert scale ranging from 1 = '*Never have done this*' to 6 = '*Almost always do this*' to indicate how often they have been doing what is described by the items to cope with work stressors. The five coping strategies are represented by three items each and a mean score ranging from 1 to 6 for each coping strategy is computed. Higher scores indicate that respondents use a specific coping strategy more often. Cronbach's alpha for the 15 items scale was acceptable in the current study (Phase 1: $\alpha = 0.84$, Phase 2 $\alpha = 0.83$), all subscales were acceptable between 0.77 and 0.92 between both phases.

**Burnout.** Burnout was assessed only in Phase 2 onwards in the wider study (after qualitative findings highlighted this outcome) using the 19-item Copenhagen Burnout Inventory (CBI) [36], which measured three different areas of burnout: personal (six items), work-related (seven items) and client-related (six items). The items (e.g., Does your work frustrate you?) were rated on a five-point Likert scale (wording differs across items) scored from 0 to 100. For each area of burnout, a mean score (ranging from 0 to 100) was calculated, with higher scores indicating greater burnout. In the current study, the burnout scores in each area are categorised into Low, Moderate, High, and Severe burnout using the cut-off scores (see Table 3) as frequently cited in the literature [37]. Cronbach's alpha was acceptable for the personal burnout scale ($\alpha = 0.90$), the work-burnout scale ($\alpha = 0.79$) and the client burnout scale ($\alpha = 0.85$).

## Data analysis

All analyses were conducted in SPSS 26 and any missing data were addressed prior to analyses. Initially, respondents who did not complete any items on one or more of the scales (SWEMWBS, WRQOL, Brief COPE, Clark's coping, Burnout), were excluded (n = 426). We then excluded participants (n = 3) who indicated their gender to be 'prefer not to say', as this would not have allowed meaningful analyses with this small subgroup to be conducted. This left a sample of 1410 participants (586 from Phase 1; 824 from Phase 2). The remaining missing data on the variables relevant to the analyses were 0.11%. The SWEMWBS, WRQOL and the coping items were treated as continuous variables and missing data on these items were estimated using the EM algorithm in SPSS as the data conformed to the Missing at Random (MAR) assumption [38]. Missing values on the demographic and work-related variables were minimal (0.04%) and they were not estimated. Instead, listwise deletion was used in the linear regression analyses. All Cronbach alphas reported refer to the healthcare sample used in this current study.

To account for the different distribution of occupations and countries across the Phase 1 and Phase 2 samples of the study, descriptive statistics for the wellbeing, quality of working life and the coping strategies were weighted by occupation and country. Weights were calculated based on published professional registrations and regional staffing figures by NHS. Frequencies and percentages describing the sample in both phases (see Table 1), the outcome burnout and the scales used in the regression analyses were unweighted. Independent samples t-tests were conducted to determine differences in the outcome measures between Phase 1 and 2 for all respondents. This was based on the assumption that the 1410 observations had some of the same respondents twice so independence was assumed for the purpose of t-tests based on the time interval. Several multiple linear regressions were conducted to examine whether demographic and work-related variables (age, gender, ethnicity, country of work, disability status, redeployment status, years of experience) and coping strategies were predictive of either mental wellbeing and work-related quality of life scores. Variables were centred before inputting the interaction terms to examine if coping strategies had an interaction between the phases.

## Results

### Descriptive statistics and preliminary analysis

The final effective healthcare sample contained 1410 respondents, 586 from Phase 1; 824 from Phase 2 (Table 1), with the sample predominately female (90.9%), and of White ethnicity (96.6%). Most respondents were in the 40–59 age group (61.4% and 56.1%), with no disability (90.8%), and had not been redeployed in their profession during the pandemic (78.9%). Nurses accounted for 164 (55.1%) of the 298 respondents who were redeployed; descriptive statistics for Phase 1 and Phase 2 are presented in Tables 2 and 3.

**Table 2. Unweighted descriptive statistics for key study variables and their comparison between Phase 1 and Phase 2 of the study.**

| Variable | Unweighted results | | |
|---|---|---|---|
| | Phase 1 (N = 586) | Phase 2 (N = 824) | Phase 1 vs. Phase 2 comparison[2] |
| | M (SD) | | p-value |
| **Wellbeing** | 21.41 (3.55) | 20.78 (3.38) | 0.001 |
| **Quality of working life** | 78.16 (15.06) | 75.56 (15.47) | 0.002 |
| **Coping strategies** | | | |
| *Active coping* | 6.02 (1.62) | 5.52 (1.62) | 0.000 |
| *Planning* | 5.81 (1.75) | 5.49 (1.76) | 0.000 |
| *Positive reframing* | 5.87 (1.58) | 5.64 (1.59) | 0.007 |
| *Acceptance* | 6.42 (1.38) | 6.16 (1.48) | 0.001 |
| *Use of emotional support* | 5.04 (1.75) | 4.98 (1.74) | 0.540 |
| *Use of instrumental support* | 4.48 (1.74) | 4.52 (1.75) | 0.620 |
| *Venting* | 3.44 (1.40) | 4.23 (1.61) | 0.000 |
| *Substance use* | 2.77 (1.41) | 2.77 (1.38) | 0.944 |
| *Behavioural disengagement* | 2.58 (1.19) | 2.86 (1.34) | 0.000 |
| *Self-blame* | 3.28 (1.65) | 3.74 (1.76) | 0.000 |
| *Family-work segmentation* | 4.99 (0.89) | 5.06 (0.93) | 0.170 |
| *Work-family segmentation* | 4.61 (1.06) | 4.57 (1.04) | 0.495 |
| *Working to improve skills/efficiency* | 4.38 (1.01) | 4.33 (1.00) | 0.359 |
| *Recreation and relaxation* | 3.68 (1.25) | 3.61 (1.25) | 0.278 |
| *Exercise* | 4.08 (1.32) | 3.89 (1.39) | 0.010 |

Note.

[1] p-value associated with independent t-tests.

**Table 3. Weighted[2] descriptive statistics for key study variables and their comparison between Phase 1 and Phase 2 of the study.**

| Variable | Weighted results[2] | | |
|---|---|---|---|
| | Phase 1 (N = 586) | Phase 2 (N = 824) | Phase 1 vs. Phase 2 comparison[2] |
| | M (SD) | | p-value |
| **Wellbeing** | 21.08 (3.41) | 20.26 (3.15) | 0.000 |
| **Quality of working life** | 77.46 (16.76) | 71.72 (15.33) | 0.000 |
| **Coping strategies** | | | |
| *Active coping* | 6.03 (1.64) | 5.48 (1.73) | 0.000 |
| *Planning* | 5.91 (1.78) | 5.55 (1.87) | 0.004 |
| *Positive reframing* | 5.84 (1.62) | 5.47 (1.58) | 0.000 |
| *Acceptance* | 6.51 (1.37) | 6.10 (1.49) | 0.000 |
| *Use of emotional support* | 5.05 (1.76) | 4.95 (1.69) | 0.369 |
| *Use of instrumental support* | 4.48 (1.84) | 4.36 (1.74) | 0.337 |
| *Venting* | 3.57 (1.43) | 4.24 (1.64) | 0.000 |
| *Substance use* | 2.87 (1.57) | 2.96 (1.53) | 0.362 |
| *Behavioural disengagement* | 2.63 (1.20) | 2.98 (1.33) | 0.000 |
| *Self-blame* | 3.56 (1.91) | 4.11 (1.85) | 0.000 |
| *Family-work segmentation* | 4.98 (0.96) | 5.13 (0.84) | 0.010 |
| *Work-family segmentation* | 4.55 (1.07) | 4.53 (1.01) | 0.696 |
| *Working to improve skills/efficiency* | 4.44 (1.03) | 4.32 (0.99) | 0.062 |
| *Recreation and relaxation* | 3.73 (1.25) | 3.41 (1.22) | 0.000 |
| *Exercise* | 4.11 (1.40) | 3.59 (1.32) | 0.000 |

Note.

[1]p-value associated with independent t-tests.

[2] The results were weighted by two-factor weighting by occupation and country.

The results showed that wellbeing and quality of working life scores for healthcare professionals were lower in Phase 2 compared to Phase 1 (p<0.001). Overall, Midwifery professionals had slightly lower wellbeing scores across both study phases (Phase 1: 21.31 (3.74), Phase 2: 20.52 (3.75) than Nurses (Phase 1: 21.79 (4.07), Phase 2: 20.72 (3.38) or AHP professionals (Phase 1: 21.28 (3.18), Phase 2: 20.84 (3.34)). Redeployment and age had significant but weak correlations with both wellbeing and WRQoL in Phases 1 and 2, but the mean differences were small. There also seemed to be lower scores for using positive approach coping strategies (active coping, planning, acceptance, positive reframing) while higher scores were evident in the use of negative avoidant coping strategies (venting, self-blame, behavioral disengagement) between Phase 1 and Phase 2 of the study.

Levels of client related-burnout were found to be much lower than personal or work-related burnout suggesting that clients are rarely the reasons for staff burnout as outlined in Table 4. Across the healthcare workforce in this current study, AHPs were found to have lower mean scores than nursing or midwifery staff in all three burnout categories (personal, work, client). In addition, we found that overall, for personal burnout, 23.9% of respondents indicated high or severe burnout levels and a further 48.1% moderate burnout levels. In relation to work-related burnout, 20.4% of responses pointed to high or severe burnout levels and a further 42.2% to moderate burnout levels. Finally, in relation to client-related burnout, 86.4% of respondents measured low levels of client burnout and only 12.0% a moderate burnout level.

**Table 4. Descriptive statistics for burnout[*].**

| Burnout | Mean (SD) | Low n (%) | Moderate n (%) | High n (%) | Severe n (%) |
|---|---|---|---|---|---|
| **Personal** | 58.82 (19.48) | 231 (28.0) | 396 (48.1) | 174 (21.1) | 23 (2.8) |
| *Nursing* | 61.74 (19.58) | 61 (21.8) | 137 (48.9) | 71 (25.4) | 11 (3.9) |
| *Midwifery* | 63.39 (16.87) | 9 (16.1) | 30 (53.6) | 16 (28.6) | 1 (1.8) |
| *AHPs* | 56.62 (19.43) | 161 (33.0) | 229 (46.9) | 87 (17.8) | 11 (2.3) |
| **Work** | 54.67 (21.16) | 308 (37.4) | 348 (42.2) | 158 (19.2) | 10 (1.2) |
| *Nursing* | 58.75 (20.49) | 85 (30.4) | 123 (43.9) | 66 (23.6) | 6 (2.1) |
| *Midwifery* | 60.46 (21.15) | 12 (21.4) | 32 (57.1) | 11 (5.7) | 1 (1.8) |
| *AHPs* | 51.67 (21.05) | 211 (43.2) | 193 (39.5) | 81 (16.6) | 3 (0.6) |
| **Client** | 25.02 (19.66) | 712 (86.4) | 99 (12.0) | 11 (1.3) | 2 (0.2) |
| *Nursing* | 26.05 (20.16) | 233 (83.2) | 45 (16.1) | 1 (0.4) | 1 (0.4) |
| *Midwifery* | 26.12 (19.68) | 47 (83.9) | 8 (14.3) | 1 (1.8) | 0 (0.0) |
| *AHPs* | 24.61 (19.38( | 432 (88.5) | 46 (9.4) | 9 (1.8) | 1 (0.2) |

[*]Only measured in Phase 2

## Independent t-tests

Results from independent t-tests demonstrated significant differences in several variables between Phase 1 and 2; SWEMWBS (t = 3.68, p<0.001,d = 0.25); total WRQoL (t = 5.48, p < .001, d = 0.36), active coping (t = 4.97, p<0.001, d = 0.33), planning (t = 2.97, p<0.01, d = 0.20), positive reframing (t = 3.50, p<0.001, d = 0.23), acceptance (t = 4.22, p<0.001, d = 0.29), venting (t = -6.69, p<0.001, d = 0.44), behavioural disengagement (t = -4.20, p<0.001. d = 0.28), self-blame (t = -4.41, p<0.001, d = 0.29), family-work segmentation (t = -2.58, p<0.01, d = 0.17), recreation and relaxation (t = 3.91, p<0.001, d = 0.26), exercise (t = 5.70, p<0.001, d = 0.38). All were significantly lower (i.e. worse) in Phase 2.

## Regression analyses

Results of the variance inflation factor ($< 10$), and collinearity tolerance ($> 0.10$) suggest that the estimated βs are well established in all regression models indicating no collinearity (Field, 2013). The unstandardized regression coefficients (b), the standardized regression coefficients (β), for the final regression models are reported in Tables 5 and 6.

In Phase 1, demographic and work-related characteristics (sex, age, ethnic background, country of work, occupational group, number of years of work experience, redeployment status and disability status) accounted for 3.5% of the variance within the wellbeing model, $F(8, 575) = 2.62$, $p < .05$. The final model as a whole explains 39.6% of the variance, this means that coping strategies account for 36.1%, $F(15, 560) = 15.95$, $p < .001$). Only six of the coping variables contributed statistically significantly to the explanation of predicting positive mental wellbeing. Positive framing (β = .18, $p < .001$), acceptance (β = .08, $p < .05$), use of emotional support (β = .10, $p < .05$), work-family segmentation (β = .09, $p < .05$), working to improve skills/efficiency (β = .12, $p < .05$). All had a positive impact, except for self-blame which showed a negative impact ($β = -.34$, $p < .001$).

Also in Phase 1, demographic and work-related characteristics (sex, age, ethnic background, country of work, occupational group, number of years of work experience, redeployment status and disability status) accounted for 3.8% of the variance within the Total WRQoL model, ($F(8, 575) = 2.88$, $p < .01$). The final model as a whole explains 32.9% of the variance, this means that coping strategies account for 29.1%, ($F(15, 560) = 11.96$, $p < .001$). Ten of the coping variables contributed significantly to the explanation of work quality of life. Active

**Table 5. Regression analysis examining coping strategies as predictors of wellbeing.**

| Predictor variable | Phase 1 (N = 586) | | | Phase 2 (N = 824) | | | Interaction between phase*coping strategies (n = 1410) |
|---|---|---|---|---|---|---|---|
| | B | β | p-value | b | β | p-value | p-value |
| Gender | 1.144 | .092 | .007 | -.022 | -.002 | .946 | .071 |
| Age | .206 | .065 | .180 | .207 | .072 | .067 | **.038** |
| Ethnicity | -.338 | -.041 | .232 | -.212 | -.026 | .349 | .127 |
| Country of work | .017 | .006 | .854 | .114 | .040 | .150 | .197 |
| Occupation | -.362 | -.085 | .018 | -.251 | -.069 | .017 | .001 |
| Redeployment | .542 | .061 | .073 | .436 | .053 | .048 | .009 |
| Experience | -.027 | -.011 | .817 | -.124 | -.053 | .177 | .311 |
| Disability | -.622 | -.052 | .127 | .301 | .009 | .726 | .464 |
| **Coping Strategies** | | | | | | | |
| Active coping | .206 | .094 | .063 | .271 | .130 | .001 | .664 |
| Planning | -.159 | -.078 | .142 | -.205 | -.107 | .009 | .595 |
| Positive reframing | .407 | .181 | .000 | .126 | .059 | .087 | **.037** |
| Acceptance | .211 | .082 | .049 | .231 | .102 | .001 | .896 |
| Use of emotional support | .221 | .109 | .013 | .416 | .214 | .000 | .063 |
| Use of instrumental support | .032 | .016 | .723 | .029 | .015 | .687 | .942 |
| Venting | .022 | .009 | .830 | -.232 | -.110 | .000 | **.041** |
| Substance use | -.126 | -.050 | .165 | -.140 | -.057 | .050 | .826 |
| Behavioural disengagement | -.193 | -.065 | .089 | -.398 | -.158 | .000 | .101 |
| Self-blame | -.737 | -.342 | .000 | -.511 | -.266 | .000 | **.034** |
| Family-work segmentation | -.129 | -.032 | .418 | -.290 | -.080 | .009 | .366 |
| Work-family segmentation | .317 | .094 | .023 | .187 | .058 | .067 | .370 |
| Working to improve skills/efficiency | .414 | .117 | .002 | .288 | .086 | .007 | .402 |
| Recreation and relaxation | .066 | .023 | .567 | .169 | .062 | .049 | .535 |
| Exercise | .086 | .032 | .396 | .126 | .052 | .081 | .641 |

*Note*. b = unstandardised estimate; β = standardised estimate. All analyses controlled for participants' sex, age, ethnic background, country of work, occupational group, number of years of work experience, and disability status.

coping (β = .14, $p < .001$), positive framing (β = .16, $p < .001$), use of instrumental support (β = .09, $p < .05$), family-work segmentation (β = .15, $p < .001$), work-family segmentation (β = .09, $p < .001$), working to improve skills/efficiency (β = .09, $p < .05$), recreation and relaxation (β = .11, $p < .05$) and exercise (β = .02, $p < .05$).). These were all positively associated with WRQoL, while planning (β = -.13, $p < .05$), acceptance (β =-.03, $p<0.001$), and substance use (β =-.04, $p < .0.01$) were negatively associated with WRQoL.

In Phase 2, demographic and work-related characteristics (sex, age, ethnic background, country of work, occupational group, number of years of work experience, redeployment status and disability status) accounted for 2.3% of the variance within the wellbeing model, ($F$(8, 575) = 2.44, $p < .05$). The final model as a whole explains 44.1% of the variance, this means that coping strategies account for 41.9%, ($F$(15, 798) = 27.36, $p < .001$). Ten of the coping variables contributed significantly to the explanation of mental wellbeing; with positive associations observed for active coping (β = .13, $p < .001$), acceptance (β = .10, $p < .001$), use of emotional support (β = .21, $p < .05$), working to improve skills/efficiency (β = .09, $p < .05$) and recreation and relaxation (β = .06, $p < .05$) whilst negative associations were found for planning (β = -.11, $p < .05$), venting (β = -.11, $p < .01$), substance use (β = -.06, $p < .05$), behavioural disengagement (β = -.16, $p < .001$), self-blame (β = -.27, $p < .001$), family-work segmentation (β = -.08, $p < .05$).

**Table 6. Regression analysis examining coping strategies as predictors of quality of working life.**

| Predictor variable | Phase 1 (N = 586) | | | Phase 2 (N = 824) | | | Interaction between phase*coping strategies (n = 1410) |
|---|---|---|---|---|---|---|---|
| | **B** | **β** | **p-value** | **b** | **β** | **p-value** | **p-value** |
| Gender | 2.871 | .054 | .127 | -1.526 | **-.029** | **.336** | .882 |
| Age | -.427 | -.032 | .534 | -.203 | **-.016** | **.716** | .331 |
| Ethnicity | -.067 | -.002 | .958 | -.007 | **.000** | **.995** | .952 |
| Country of work | -.967 | -.088 | .017 | .509 | **.039** | **.196** | .482 |
| Occupation | .934 | .051 | .169 | 1.225 | **.074** | **.018** | **.003** |
| Redeployment | 3.329 | .089 | .014 | 1.943 | **.052** | **.075** | **.003** |
| Experience | 2.04 | .020 | .695 | .121 | **.011** | **.791** | .538 |
| Disability | -.384 | -.008 | .833 | 3.324 | **.065** | **.026** | .110 |
| **Coping Strategies** | | | | | | | |
| Active coping | 1.308 | .141 | .000 | .933 | .098 | .026 | .594 |
| Planning | -1.154 | -.134 | .008 | -1.431 | -.163 | .000 | .571 |
| Positive reframing | 1.555 | .163 | .017 | .378 | .039 | .299 | .072 |
| Acceptance | -.330 | -.030 | .001 | .479 | .046 | .178 | .213 |
| Use of emotional support | .837 | .097 | .491 | 1.392 | .157 | .000 | .291 |
| Use of instrumental support | .371 | .044 | .034 | .371 | .042 | .295 | .909 |
| Venting | -1.203 | -.112 | .352 | -1.146 | -.119 | .000 | .891 |
| Substance use | -.457 | -.043 | .010 | .205 | .018 | .562 | .241 |
| Behavioural disengagement | -.715 | -.057 | .259 | -2.210 | -.192 | .000 | **.019** |
| Self-blame | -2.463 | -.270 | .158 | -1.670 | -.190 | .000 | .165 |
| Family-work segmentation | -2.474 | -.146 | .000 | -2.071 | -.124 | .000 | .679 |
| Work-family segmentation | 1.288 | .090 | .001 | 1.644 | .111 | .001 | .643 |
| Working to improve skills/efficiency | 1.333 | .089 | .038 | 1.307 | .085 | .014 | .906 |
| Recreation and relaxation | 1.269 | .105 | .027 | .816 | .066 | .055 | .439 |
| Exercise | -.167 | -.015 | .013 | .405 | .037 | .259 | .291 |

*Note*. b = unstandardised estimate; β = standardised estimate. All analyses controlled for participants' country of work, occupational group, number of years of work experience, sex, age, disability status and ethnic background.

Also in Phase 2, for total WRQol, demographic and work-related characteristics (sex, age, ethnic background, country of work, occupational group, number of years of work experience, redeployment status and disability status) accounted for 3.6% of the variance within the model, ($F_{(8, 813)} = 3.81$, $p < .001$). The final model as a whole explains 34.2% of the variance, this means that coping strategies account for 30.6%, ($F_{(15, 821)} = 18.02$, $p < .001$). Nine of the coping variables contributed significantly to the explanation of quality of working life though not the same strategies that were predictive in Phase 1. Active coping ($\beta = .10$, $p < .05$), use of emotional support ($\beta = .16$, $p < .001$), work-family segmentation ($\beta = .11$, $p < .001$), and working to improve skills/efficiency ($\beta = .09$, $p < .05$) were significantly positive, while planning ($\beta = -.16$, $p < .05$), venting ($\beta = -.12$, $p < .001$), self-blame ($\beta = -.19$, $p < .001$), family-work segmentation ($\beta = -.12$, $p < .001$) were significantly negative.

## Discussion

### Summary of findings and comparison with other literature

The current study compared cross-sectional data collected from healthcare staff (nurses, midwives and AHPs) in the UK at two-time points during the COVID-19 pandemic; Phase 1 of

the study (7[th] May–3[rd] July 2020) and Phase 2 of the study (17[th] November 2020–1[st] February 2021). The results showed that wellbeing and quality of working life for healthcare professionals were significantly lower in Phase 2 compared to Phase 1. Mean Wellbeing scores are similar to another pandemic-era study by Smith et al. [39] using the same wellbeing measure (SWEMWBS) which reported similar mean scores among UK-based respondents of all occupations 20.8-SD 5.1; compared to 21.08-SD 3.4 (Phase 1) and 20.26- SD 3.1 (Phase 2) in the current study. However, a study by Firat et al. [40] reported a mean score of 25.01 (SD: 5.44) in healthcare personnel during COVID-19 in Turkey on the same scale which is higher than the findings of this current study. Prior to the pandemic, Durkin et al. [41] reported a mean score of 25.2 (3.1) amongst UK community nurses. Furthermore, normative (population norms) level of wellbeing using SWEMWBS have previously been reported as a mean score of 23.6 [31, 42], meaning that the wellbeing of the healthcare workers in this study were 3 points below the pre-COVID time.

In this current study, quality of working life decreased from 77.46 (16.76) to 71.72 (15.33) between the two phases. These scores were lower than a Spanish pilot study involving the WRQoL scale, which reported scores of 78.13 (19.89) in nurses [43] while another study of healthcare professionals in Vietnam reported scores of 95.52 and 92.10 [44]. However, the scores of this current study for WRQoL were higher than studies in Iran which reported scores of 68.81 (19.12) for nurses caring for patients with Covid-19 in public hospitals' wards [45] and 50.64 (11.55) in nurses working in Tehran University of Medical Sciences Hospitals [46].

When coping was added to the model as an interaction with Phase, the effects of the individual study Phase disappeared for many of the variables, suggesting that coping explains the difference and is important for the healthcare workforce. Our results indicated that coping strategies can be a critical component in overall wellbeing and quality of working life, highlighting that, as the pandemic continued, more negative avoidant coping strategies were utilized. Pre-Covid evidence supported the use of coping strategies to help reduce stressors, regulate emotional and behavioral responses to improve psychological wellbeing and quality of working life [47–55]. Within this current study, respondents used similar positive coping strategies between Phases 1 and 2 such as active coping. This result was in line with literature which suggests that social support through instrumental and emotional support alongside positive attitude and active coping are important coping mechanisms that can increase the resilience of the healthcare workforce while having a positive impact on quality of working life and wellbeing [48, 51, 54, 56–59]. These findings support the present study findings which suggest that positive approach coping strategies (active coping, planning, acceptance, positive reframing, social support) are positively associated with wellbeing and work-related quality of life. In contrast, an Italian survey examining the psychological effects of the COVID-19 pandemic among healthcare professionals found that social support negatively impacted wellbeing as higher levels of social support were associated with high levels of stress [60].

Findings in this current study highlighted a decrease in the use of positive coping strategies (active coping, planning, acceptance, positive reframing) while a significant increase was evident in the use of negative avoidant coping strategies (venting, self-blame, behavioral disengagement) between Phases 1 and 2. Similarly, Flesia et al. [57] and Babore et al. [60] have reported that higher usage of avoidant type strategies negatively impacted psychological state. From Japan, Tahara et al. [61] highlighted that while job satisfaction could be a resilience factor, decreased resilience and poorer mental state during the COVID-19 pandemic have occurred and this potentially increases the use of more negative, avoidant type coping strategies by the healthcare workforce. These authors found over 70 per cent of their survey respondents reported using avoidant type strategies, comparable to the findings of this present study. Furthermore, the current study adds additional insight into the role of coping by virtue of

having observations at two separate timepoints in the course of the pandemic. For wellbeing, the association with using positive reframing greatly increased from the summer period of 2020 (Phase 1) to the winter period 2020/21 (Phase 2), suggesting this type of coping may have become more impactful as the pandemic wore on. On the other hand, the impact of venting, and self-blame as coping strategies greatly decreased wellbeing over the same period. For work-related quality of life, similar to wellbeing, the association with using emotional support greatly increased from Phase 1 to Phase 2, suggesting that this coping strategy could have more of an influence in positive wellbeing as the pandemic continued. The association of using behavioural engagement and planning greatly decreased from Phase 1 to Phase 2, which suggested a negative impact with using this strategy. As stress and coping strategies are interlinked, the continued uncertainty and increasing stressors associated with the pandemic could have further detrimental effects on the wellbeing and work-related quality of life of healthcare professionals.

Globally, as the pandemic continues, as the virus mutates, with varied vaccine rollout and rising death tolls in many countries, the OVID-19 pandemic is feared to be leading to a severe, long-lasting psychological impact on healthcare workers [23, 49, 52, 58, 62, 63]. As highlighted within this study, healthcare workers are beginning to feel burnt out, with over 60 per cent of our respondents experiencing moderate to severe levels of burnout between November 2020-January 2021. Similar to this study, a longitudinal study across the UK and Ireland found that the level of burnout was negatively associated with wellbeing and, as burnout increased, wellbeing scores were lower [64]. This has also been acknowledged by others who reported that even as restrictions ease, the healthcare workforce continues to operate under pressure increasing the risk of burnout particularly among those involved directly in patient care [63, 65–70].

Nishimura et al. [69] acknowledged that the pandemic has altered the world that we live and work in, which makes the deterioration of wellbeing and quality of working life and the increase in burnout of healthcare professionals concerning. While the UK had established support wellbeing centers and services during the first wave of COVID-19 for most of the NHS workforce [49], stressors continue to increase due to uncertainty and increased job demands. The decrease in positive strategies could be an explanation for the deterioration of wellbeing and quality of working life within this current study. However, another explanation could be the lack of sustainable support and services available to help this workforce deal with the heavy toll and the pandemic. Also healthcare staff with increased demands on their time, may not have space in the working day to avail of supports and when taking a day off or on leave may not want to return to or connect with the workplace in order to access wellbeing and support services. This could affect future services and the mental health of healthcare professionals.

In addition to support and wellbeing services, here we highlight Human Resource Management (HRM) practices as implemented by employers and line managers. HRM practices include training, (career) development, opportunities to engage in decision making and communication amongst staff and with supervisors. A study investigating these HRM practices in the NHS showed that HRM practices were positively related to work engagement amongst staff, which was again positively related to positive outcomes such as safety and quality of care [71]. However, current research suggests that HRM practices need to be adapted to account for unforeseen crises such as COVID-19, which might affect worker wellbeing. Evidence from the hospitality industry indicates that flexible and employee-centered HRM practices are more effective during a crisis and that those HR practices supporting work-life balance appear to be more important to manage and maintain employee wellbeing [72]. The role of line managers, orsupervisors, in implementing such HRM policies is of special relevance given the importance of positive relationships and social support in managing employee wellbeing [73].

However, research indicates that HRM strategies are not always converted into HRM practices and that line managers do not always implement HR practices as intended [74, 75]. This is often due to lack of awareness of policies and/or lack of importance placed on policy implementation. This suggests that line managers need adequate training themselves, and need the knowledge, skills and time to engage with their colleagues to garner the positive effects from HRM on the one hand and from social relationships on the other.

## Limitations and strengths

A major strength of the study was the use of validated scales to assess coping, wellbeing and quality of working life. Data were collected during the first and second wave of the COVID-19 pandemic in the UK and therefore provide information as to how the protracted and ever-unfolding nature of the crisis affected frontline healthcare professionals. Given the unpredictable nature of the pandemic, this current study is important in exploring the difference in wellbeing and quality of working life over the two different time points and has demonstrated how positive coping strategies and support could potentially help improve workforce wellbeing. This is a strength of the cross-sectional design of the study which allowed for an assessment of the association between the outcomes [76].

The survey is however limited by the use of the cross-sectional data collected, meaning that it is only reflective of the healthcare workforce at that current time point. Therefore, the data cannot reveal causal relationships as we cannot infer cause and effect [76, 77]. A longitudinal data collection and examination of the effects on the variables over time would enable a more detailed exploration of the relationships, however given the nature of the pandemic it would be more difficult to record this information over a long period. Data were collected online using a snowball sampling method through extensive sharing of a survey link. While this was a pragmatic way to safely gather data from a large sample during the COVID-19 pandemic, it may increase the risk of selection bias in that the sample may be over-representative either of workers with sufficient time to complete a survey, or those concerned about the wellbeing of themselves and colleagues and therefore more motivated to register their views [78–80]. This is borne out in part by the under-representation of particular groups within the workforce, such as Midwives.

The study sample had an overall representation of female respondents (overall: 90.9%, Phase 1: 91.0% and Phase 2 (9.8%), which is reflective of the composition of this part of the healthcare workforce [81–83]. However, a limitation is that there is insufficient statistical power for granular sub-group analysis, for example to examine different patterns of coping between male and female healthcare workers. Similarly, there was an over-representation of respondents who identified as being of White ethnicity (96.6%) which is higher than the UK NHS workforce which is 77.9% White ethnicity [84]. Recruiting a sample which ethnically resembles the underlying workforce has proven a challenge, which could be linked to the sampling technique utilised. Over 50 percent of the sample were AHPs, therefore not representative of the whole healthcare workforce in the UK making generalisations more challenging. The research team mitigated this limitation by weighting the data during the statistical analysis which allows for a more accurate representation of the population being examined by diminishing the effects of inherent biases of the survey. We did not recruit medical professionals as these have been the focus of others' research.

## Implications

While several wellbeing initiatives were implemented for healthcare staff during the COVID-19 pandemic, it appears from the findings of this current study that more services and/or support may help prevent a further deterioration of wellbeing and quality of working life. Indeed,

as healthcare services are rebuilt and resume, staff need time to rest and recover which is particularly challenging given the extensive patient waiting lists in the UK.

The findings support the concept that employee support across multiple levels (individual, organizational and policy level) must be implemented or sustained to establish good working conditions to support the wellbeing and quality of working life of healthcare professionals. However, staff need support (including time) and encouragement to access and use the services that are available. Blake et al. [49] proposed that wellbeing and support services should help mitigate the psychological impact of COVID-19 on the healthcare workforce. The potential of support services within the workplace are evidenced in this current study and formed part of the 15 'Good Practice Recommendations' in the Health and Social Care Workforce Phase 1 and Phase 2 main study reports [85, 86]. Managers and employing organizations need to establish or promote good work-life balance for employees by providing flexible working hours and location, regular breaks and encouraging staff to take annual leave. This, in combination with good two-way communication, may provide employees with a voice while creating a positive work environment that can sustain quality of working life and wellbeing.

## Conclusion

The COVID-19 pandemic has had a significant effect on the wellbeing, quality of working life and coping of Nurses, Midwives and AHPs across the UK as shown in the findings of this present study. While coping strategies are associated with both wellbeing and quality of working life, respondents demonstrated an increase in negative coping strategies to deal with the escalation of work pressures. Therefore, strategies must be implemented across multiple levels to help staff use positive coping strategies as protective factors for the healthcare workforce.

## Acknowledgments

The authors thank all participants for responding to these surveys during an extremely pressurised working period. Moreover, thanks to all who promoted the study including Community Care ©, Northern Ireland Practice and Education Council for Nursing and Midwifery, Royal College of Nursing, Royal College of Midwifery, Royal College of Occupational Therapists, British Dietetic Association, College of Podiatry, the five Health and Social Care Trusts in Northern Ireland, and the NISCC..

## Author Contributions

**Conceptualization:** Patricia Gillen, Ruth D. Neill, Paula McFadden.

**Data curation:** Patricia Gillen, Ruth D. Neill, Paula McFadden.

**Formal analysis:** Ruth D. Neill, John Mallett, John Moriarty.

**Funding acquisition:** Patricia Gillen, Paula McFadden.

**Investigation:** Patricia Gillen, Paula McFadden.

**Methodology:** Patricia Gillen, Ruth D. Neill, Paula McFadden.

**Project administration:** Patricia Gillen, Paula McFadden.

**Supervision:** Patricia Gillen.

**Writing – original draft:** Patricia Gillen, Ruth D. Neill, John Mallett, John Moriarty, Jill Manthorpe, Heike Schroder, Denise Currie, Susan McGrory, Patricia Nicholl, Jermaine Ravalier, Paula McFadden.

**Writing – review & editing:** Patricia Gillen, Ruth D. Neill, John Mallett, John Moriarty, Jill Manthorpe, Heike Schroder, Denise Currie, Susan McGrory, Patricia Nicholl, Jermaine Ravalier, Paula McFadden.

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
