## [Decision Letter · Decision Letter 0]

13 Sep 2021

PONE-D-21-21515Wellbeing and Coping of UK Nurses, Midwives and Allied Health Professionals during COVID-19-a cross sectional study.PLOS ONE

Dear
Dr. Gillen
,

Thank you for submitting your manuscript to PLOS ONE. After careful consideration, we feel that it has merit but does not fully meet PLOS ONE’s publication criteria as it currently stands. Therefore, we invite you to submit a revised version of the manuscript that addresses the points raised during the review process.

Please address the methodological issues raised by Reviewer 2. Pay particular attention to the presentation of the regression analysis.

We look forward to receiving your revised manuscript.

Kind regards,

Rosemary Frey

Academic Editor

PLOS ONE

Journal Requirements:

Reviewers' comments:

Reviewer's Responses to Questions

**Comments to the Author**

1. Is the manuscript technically sound, and do the data support the conclusions?

Reviewer #1: Yes

Reviewer #2: Yes

2. Has the statistical analysis been performed appropriately and rigorously? 

Reviewer #1: Yes

Reviewer #2: Yes

3. Have the authors made all data underlying the findings in their manuscript fully available?

Reviewer #1: Yes

Reviewer #2: Yes

4. Is the manuscript presented in an intelligible fashion and written in standard English?

Reviewer #1: Yes

Reviewer #2: Yes

5. Review Comments to the Author

Reviewer #1: This submission is obviously relevant and timely. It is clearly-written, and clearly-reasoned throughout. The descriptive and regression statistical analyses are competently done and well elaborated in the text and tables. The chosen measures are psychometrically sound (adequate scale alphas etc.), and appropriate to the task.

A few (very minor) suggested revisions/rephrasings...

In table 2, with both unweighted and weighted results, there is too much crammed into the one table. Is it absolutely necessary to list both unweighted and weighted here? If it is, perhaps do this as two separate tables.

367 coping strategies account for 36.1%, F(15, 560)=15.95, p<.001). Only six of the coping variables

368 contributed significantly to the explanation of mental wellbeing. Positive framing

((Rephrase, clarifying what is meant by 'significantly'. It's not clear whether this refers to importance or statistical significance.))

520 quality of working life over the two different time points and has identified//demonstrated how positive coping

521 strategies and support can//could potentially help improve workforce wellbeing. This is a strength of the cross

((I would suggest rephrasing identified/demonstrated and can/could potentially))

Reviewer #2: Thank you for giving me the pleasure to review your manuscript. Overall, it is a very nice work. However, I have the following comments

1- My concern is how the authors made sure that the 2nd wave of data collection had the same participants as in wave 1 given that other than emails , social media advertising was used and an e-survey was used for data collection?

2- There is no enough description about the mental wellbeing scale used in the study? Giving some examples of the items would be beneficial

3- The regression tables do not present the full model as sociodemographic variables were excluded from the tables although they were used as predictors.

4- As the regression analyses focused on coping strategies as predictors for mental well-being and quality of working life, this should be reflected in the results section of the abstract as well and in the introduction by reporting studies that investigated coping strategies used by healthcare workers during the pandemic.

5- Was wondering if you thought about using coping strategies as a moderator/mediator between (mental well-being and quality of working life) and burnout as an outcome

6. PLOS authors have the option to publish the peer review history of their article (what does this mean?). If published, this will include your full peer review and any attached files.

Reviewer #1: No

Reviewer #2: No

---

## [Author Response · Author response to Decision Letter 0]

6 Oct 2021

Response to Reviewers:

The authors would like to thank the reviewers for taking their time to consider this manuscript and for providing detailed feedback.

Reviewers' comments:

Reviewer #1: This submission is obviously relevant and timely. It is clearly-written, and clearly-reasoned throughout. The descriptive and regression statistical analyses are competently done and well elaborated in the text and tables. The chosen measures are psychometrically sound (adequate scale alphas etc.), and appropriate to the task.

A few (very minor) suggested revisions/rephrasings:

In table 2, with both unweighted and weighted results, there is too much crammed into the one table. Is it absolutely necessary to list both unweighted and weighted here? If it is, perhaps do this as two separate tables.

Author comments: The authors have separated the table into two (Table 2 & 3; lines-300-313).

367 coping strategies account for 36.1%, F(15, 560)=15.95, p<.001). Only six of the coping variables

368 contributed significantly to the explanation of mental wellbeing. Positive framing

((Rephrase, clarifying what is meant by 'significantly'. It's not clear whether this refers to importance or statistical significance.))

Author comments: The authors have added the following: “Only six of the coping variables contributed statistically significantly to the explanation of predicting positive mental wellbeing.”(line 357-358)

520 quality of working life over the two different time points and has identified//demonstrated how positive coping

521 strategies and support can//could potentially help improve workforce wellbeing. This is a strength of the cross

((I would suggest rephrasing identified/demonstrated and can/could potentially))

Author comments: The authors have taken the reviewers suggestions for rephrasing (lines 511-512)

Reviewer #2: Thank you for giving me the pleasure to review your manuscript. Overall, it is a very nice work. However, I have the following comments

1- My concern is how the authors made sure that the 2nd wave of data collection had the same participants as in wave 1 given that other than emails, social media advertising was used and an e-survey was used for data collection?

Author comments: The authors noted that the study was cross-sectional and have highlighted this within the limitations section of the discussion; “The survey is however limited by the use of the cross-sectional data collected, meaning that it is only reflective of the healthcare workforce at that current time point. Therefore, the data cannot reveal causal relationships as we cannot infer cause and effect”. (lines 515-517) Therefore due to confidentiality and different survey collection points we cannot ascertain which respondents answered the Phase 1 and Phase 2 surveys. The surveys are cross-sectional and stand-alone as there are different questions also asked in each, so it cannot be determined if the same participants completed the survey. Additionally, the same methods of data collection were used in both phases and the same data recruitment methods as noted on lines 136-148.

2- There is not enough description about the mental wellbeing scale used in the study? Giving some examples of the items would be beneficial

Author comments: Lines 180-184 have been amended to include: “Mental wellbeing was assessed using the short version of the Short Warwick-Edinburgh Mental Wellbeing Scale (SWEMWBS) [29], a positively worded seven-item scale assessing statements about thoughts and feeling which asks respondents to describe their experiences of how often they felt this way in the last two weeks.” This statement is a reflective description of the scale as detailed by the creators of the SWEMWS, Stewart et al. (2009).

3- The regression tables do not present the full model as sociodemographic variables were excluded from the tables although they were used as predictors.

Author comments: The authors have amended Tables 5 and 6 to include the sociodemographic variables (lines 341-347)

4- As the regression analyses focused on coping strategies as predictors for mental well-being and quality of working life, this should be reflected in the results section of the abstract as well and in the introduction by reporting studies that investigated coping strategies used by healthcare workers during the pandemic.

Author comments: The authors have amended the abstract to include the following statement: “The COVID-19 pandemic had a significant effect on the psychological wellbeing, quality of working life which decreased while the use of negative coping and burnout of these healthcare professionals increased.”(lines 44-45).

5- Was wondering if you thought about using coping strategies as a moderator/mediator between (mental well-being and quality of working life) and burnout as an outcome

Author comments: The authors did consider this but as burnout was only measured in Phase 2, a comparison could not be made. The authors decided to use coping strategies as a moderator between mental wellbeing, quality of working life and burnout in a later study comparing Phases 1, 2 and the recently completed Phase 3 data.

As per the Academic editor’s email, please amend the financial disclosure statement to reflect what is in the manuscript (lines565-571):

Funding Statement: The authors would like to thank the Northern Ireland Social Care Council (NISCC), and the Southern Health and Social Care Trust in Northern Ireland for seed funding and the Public Health Agency R&D Office Northern Ireland for study funding and supported by the National Institute for Health Research (NIHR) Policy Research Programme, through the Policy Research Unit in Health and Social Care Workforce, PR-PRU-1217-21202. The views expressed are those of the authors and not necessarily those of the funder or the NIHR Policy Research Programme.

---

## [Decision Letter · Decision Letter 1]

4 May 2022

PONE-D-21-21515R1Wellbeing and Coping of UK Nurses, Midwives and Allied Health Professionals during COVID-19-a cross sectional study.PLOS ONE

Dear Dr. Gillen,

Thank you for submitting your manuscript to PLOS ONE. After careful consideration, we feel that it has merit but does not fully meet PLOS ONE’s publication criteria as it currently stands. Therefore, we invite you to submit a revised version of the manuscript that addresses the points raised during the review process.

In addition to the reviewer comments, please address the followingPlease specify whether the design is cross-sectional or longitudinal given that data was collected at two time points.

Please explain how the sample was selected for both surveys- was this a random sample? If the sample is random, the study design would be more correctly described as a repeated cross-sectional design.

How did you address the possibility of a cohort effect?

Please provide the rationale for including burnout only in Phase Two?

Please specify the type of multiple regression that was conducted.

Independent t-tests

How did you address the chance that you will make a Type I error?

Please submit your revised manuscript by  Jun 18 2022 11:59PM. If you will need more time than this to complete your revisions, please reply to this message or contact the journal office at plosone@plos.org. Please include the following items when submitting your revised manuscript:A rebuttal letter that responds to each point raised by the academic editor and reviewer(s). You should upload this letter as a separate file labeled 'Response to Reviewers'.A marked-up copy of your manuscript that highlights changes made to the original version. You should upload this as a separate file labeled 'Revised Manuscript with Track Changes'.An unmarked version of your revised paper without tracked changes. You should upload this as a separate file labeled 'Manuscript'.If applicable, we recommend that you deposit your laboratory protocols in protocols.io to enhance the reproducibility of your results. Protocols.io assigns your protocol its own identifier (DOI) so that it can be cited independently in the future. For instructions see: https://journals.plos.org/plosone/s/submission-guidelines#loc-laboratory-protocols. Additionally, PLOS ONE offers an option for publishing peer-reviewed Lab Protocol articles, which describe protocols hosted on protocols.io. Read more information on sharing protocols at https://plos.org/protocols?utm_medium=editorial-email&utm_source=authorletters&utm_campaign=protocols.

We look forward to receiving your revised manuscript.

Kind regards,

Rosemary Frey

Academic Editor

PLOS ONE

Journal Requirements:

Reviewers' comments:

Reviewer's Responses to Questions

**Comments to the Author**

1. If the authors have adequately addressed your comments raised in a previous round of review and you feel that this manuscript is now acceptable for publication, you may indicate that here to bypass the “Comments to the Author” section, enter your conflict of interest statement in the “Confidential to Editor” section, and submit your "Accept" recommendation.

Reviewer #2: (No Response)

Reviewer #3: (No Response)

2. Is the manuscript technically sound, and do the data support the conclusions?

Reviewer #2: No

Reviewer #3: Yes

3. Has the statistical analysis been performed appropriately and rigorously? 

Reviewer #2: Yes

Reviewer #3: Yes

4. Have the authors made all data underlying the findings in their manuscript fully available?

Reviewer #2: Yes

Reviewer #3: Yes

5. Is the manuscript presented in an intelligible fashion and written in standard English?

Reviewer #2: Yes

Reviewer #3: Yes

6. Review Comments to the Author

Reviewer #2: Thank you for giving me the opportunity to review this manuscript. It was obvious that a lot of work has been committed to it. However, I have the following comments

Abstract

1- The authors mentioned that their data is cross-sectional and then they mentioned that they collected data at two-time points, and this is confusing since cross-sectional means data are collected at one time period of the study.

Introduction

1- Well-written and culturally situated

2- In the study aim, the authors did not mention they also studied burnout among their sample, but in the abstract section, it was mentioned. Please be consistent

Design:

1- Can you please detail how the emails were obtained for participants?

2- Can you give an example or two of some of the SWEMWBS scale items?

3- Why burnout was only measured in the 2nd phase of the study?

Results:

1- I’m not sure what is the point of comparing the two phases of data collection. I guess the authors may be right in that some respondents of phase one are the same in phase two, but this is just an assumption. Therefore, to say for example, quality of working life was lower in phase 2 than phase 1, this may relate to that the participants are different , not to the variable per se.

2- What type of regression analysis was used?

Discussion

Saying that coping was used as an interaction with phase will let the reader understands that an interaction term was created, but this is not the case in the study. Therefore, please change the wording

- The authors mentioned “Findings in this current study highlighted a decrease in the use of positive coping strategies 430 (active coping, planning, acceptance, positive reframing) while a significant increase was evident 431 in the use of negative avoidant coping strategies (venting, self-blame, behavioral disengagement) 23 432 between Phases 1 and 2”. I am still having hard time to grasp comparing the variables between phase 1 and 2 as I mentioned before, this may relate to having a different sample in phase 2 from phase 1

- “The association of using behavioural engagement and planning greatly decreased from Phase 1 to Phase 2, which suggested a negative impact with using this strategy. As stress and coping strategies are interlinked, the continued uncertainty and increasing stressors associated with the pandemic could 450 have further detrimental effects on the wellbeing and work-related quality of life of healthcare professionals.” Comparing this paragraph with the results in the regression table, shows that planning was not a significant predictor of well-being in phase 1, but it was significant for phase 2. Then planning has been increased by looking at beta values . The same for behavioral disengagement.

I recommend that the authors just pick one phase of the study to report the findings and make sure that reporting and discussing the findings are congruent with the numbers in the tables.

Reviewer #3: This is a very interesting study. The article is well organised and written and the authors have addressed the points raised by the other reviewers.

7. PLOS authors have the option to publish the peer review history of their article (what does this mean?). If published, this will include your full peer review and any attached files.

Reviewer #2: No

Reviewer #3: **Yes: **Andrew Paul Smith

---

## [Author Response · Author response to Decision Letter 1]

5 Jul 2022

12th May 2022

Wellbeing and Coping of UK Nurses, Midwives and Allied Health Professionals during COVID-19-a cross sectional study

Response to reviewers comments:

The authors would like to thank the reviewers for taking their time to consider this manuscript and for providing detailed feedback.

In addition to the reviewer comments, please address the following

Please specify whether the design is cross-sectional or longitudinal given that data was collected at two time points.

Authors’ comments: The authors would like to note that the abstract and methods state that cross-sectional data was collected at two different time points during the pandemic.

Please explain how the sample was selected for both surveys- was this a random sample? If the sample is random, the study design would be more correctly described as a repeated cross-sectional design.

Authors’ comments: The authors have now noted that a similar cross-sectional survey was repeated at different time points through a convenience sampling recruitment approach.

How did you address the possibility of a cohort effect?

Authors’ comments: To mitigate a cohort effect as mentioned by the authors in the analysis section of the manuscript, data was weighted. The authors also noted in the limitations that “a longitudinal data collection and examination of the effects on the variables over time would enable a more detailed exploration of the relationships, however given the nature of the pandemic it would be more difficult to record this information over a long period.” 

Please provide the rationale for including burnout only in Phase Two?

Authors’ comments: The authors introduced the burnout scale into Phase 2 following on from qualitative data presented in Phase 1 of the study. The scale has since been used in the subsequent phases of the wider study (Phases 3 and 4) and is relevant to the current academic literature citing burnout amongst health and social care professionals.

Please specify the type of multiple regression that was conducted.

Authors’ comments: The authors’ have now noted that a linear regression was used.

Independent t-tests - The Independent Samples t Test compares the means of two independent groups in order to determine whether there is statistical evidence that the associated population means are significantly different. The Independent Samples t Test is a parametric test. 

How did you address the chance that you will make a Type I error? 

Authors’ comments: The authors’ looked to decreased the chance of Type 1 errors by adjusting the alpha level. The authors did use significant levels of 0.05 and 0.01, however issues which lower the alpha level can lead to the results of the hypothesis test not capturing the true difference in the data.

Reviewers' comments:

Reviewer's Responses to Questions 

Comments to the Author

Reviewer #2: Thank you for giving me the opportunity to review this manuscript. It was obvious that a lot of work has been committed to it. However, I have the following comments

Abstract

1- The authors mentioned that their data is cross-sectional and then they mentioned that they collected data at two-time points, and this is confusing since cross-sectional means data are collected at one time period of the study.

Authors’ comments: The authors have now mentioned that this is a wider study which utilised a repeated cross-sectional design, and the data presented in this current paper were collected at two different time periods; May-July 2020 (Phase 1 of study) and November-February 2021 (Phase 2 of study).

Introduction

1- Well-written and culturally situated

2- In the study aim, the authors did not mention they also studied burnout among their sample, but in the abstract section, it was mentioned. Please be consistent

Authors’ comments: The authors have added in this into the study aims for consistency.

Design:

1- Can you please detail how the emails were obtained for participants?

Authors’ comments: The authors did not obtain any emails from the participants, no personal identifying information was obtained. “Respondents were recruited by convenience sampling through emails, newsletters and social media posts of employers, regulatory bodies, professional communications, professional associations and workplace unions. Participation in the study at each time point was voluntary and the data were collected through an online survey hosted on the Qualtrics platform.” Participants were not asked for email address, therefore while we may have recruited the same participants at each cross-sectional study we cannot confirm this.

2- Can you give an example or two of some of the SWEMWBS scale items?

Authors’ comments: Examples of the scale items are: I’ve been dealing with problems well, I’ve been feeling relaxed.

3- Why burnout was only measured in the 2nd phase of the study?

Authors’ comments: The authors introduced the burnout scale into Phase 2 following on from qualitative data presented in Phase 1 of the study. The scale has since been used in the subsequent phases of the wider study (Phases 3 and 4) and is relevant to the current academic literature citing burnout amongst health and social care professionals.

Results:

1- I’m not sure what is the point of comparing the two phases of data collection. I guess the authors may be right in that some respondents of phase one are the same in phase two, but this is just an assumption. Therefore, to say for example, quality of working life was lower in phase 2 than phase 1, this may relate to that the participants are different , not to the variable per se.

Authors’ comments: The authors’ note in the limitations section the issue with cross-section design and acknowledge that comparing two different dataset may not have had the same participants but this allows a comparison to a generalised overview of the possible trend.

2- What type of regression analysis was used?

Authors’ comments: The authors note that linear regressions were used and have added this to the manuscript.

Discussion

Saying that coping was used as an interaction with phase will let the reader understands that an interaction term was created, but this is not the case in the study. Therefore, please change the wording.

Authors’ comments: The authors note in the analysis that interaction terms were created.

- The authors mentioned “Findings in this current study highlighted a decrease in the use of positive coping strategies 430 (active coping, planning, acceptance, positive reframing) while a significant increase was evident 431 in the use of negative avoidant coping strategies (venting, self-blame, behavioral disengagement) 23 432 between Phases 1 and 2”. I am still having hard time to grasp comparing the variables between phase 1 and 2 as I mentioned before, this may relate to having a different sample in phase 2 from phase 1

Authors’ comments: The authors acknowledge generalisation cannot be made but found it important to examine the variables over multiple time periods of the COVID-19 pandemic to provide an example or note a possible trend in issues affecting this workforce during a difficult time.

- “The association of using behavioural engagement and planning greatly decreased from Phase 1 to Phase 2, which suggested a negative impact with using this strategy. As stress and coping strategies are interlinked, the continued uncertainty and increasing stressors associated with the pandemic could 450 have further detrimental effects on the wellbeing and work-related quality of life of healthcare professionals.” Comparing this paragraph with the results in the regression table, shows that planning was not a significant predictor of well-being in phase 1, but it was significant for phase 

2. Then planning has been increased by looking at beta values . The same for behavioral disengagement.

Authors’ comments: The authors’ have negative beta values for the planning and behavioural disengagement.

I recommend that the authors just pick one phase of the study to report the findings and make sure that reporting and discussing the findings are congruent with the numbers in the tables.

Authors’ comments: The authors acknowledge this statement but do feel that this paper will add something to the current literature and is important to explore across the two phases. However if the reviewer feels strongly we can change this.

Reviewer #3: This is a very interesting study. The article is well organised and written and the authors have addressed the points raised by the other reviewers.

---

## [Decision Letter · Decision Letter 2]

22 Aug 2022

Wellbeing and Coping of UK Nurses, Midwives and Allied Health Professionals during COVID-19-a cross sectional study.

PONE-D-21-21515R2

Dear Dr. Gillen,

We’re pleased to inform you that your manuscript has been judged scientifically suitable for publication and will be formally accepted for publication once it meets all outstanding technical requirements.

Kind regards,

Rosemary Frey

Academic Editor

PLOS ONE

Additional Editor Comments (optional):

Reviewers' comments:

Reviewer's Responses to Questions

**Comments to the Author**

1. If the authors have adequately addressed your comments raised in a previous round of review and you feel that this manuscript is now acceptable for publication, you may indicate that here to bypass the “Comments to the Author” section, enter your conflict of interest statement in the “Confidential to Editor” section, and submit your "Accept" recommendation.

Reviewer #2: All comments have been addressed

Reviewer #3: All comments have been addressed

2. Is the manuscript technically sound, and do the data support the conclusions?

Reviewer #2: Yes

Reviewer #3: Yes

3. Has the statistical analysis been performed appropriately and rigorously? 

Reviewer #2: Yes

Reviewer #3: Yes

4. Have the authors made all data underlying the findings in their manuscript fully available?

Reviewer #2: Yes

Reviewer #3: Yes

5. Is the manuscript presented in an intelligible fashion and written in standard English?

Reviewer #2: Yes

Reviewer #3: Yes

6. Review Comments to the Author

Reviewer #2: No further comments, the manuscript is now improved and eligible for publication. The authors satisfactorily responded to all comments.

Reviewer #3: The authors have addressed the issues raised by the reviewers and the manuscript is now in a form that is suitable for publication.

7. PLOS authors have the option to publish the peer review history of their article (what does this mean?). If published, this will include your full peer review and any attached files.

Reviewer #2: **Yes: **Ghada Shahrour

Reviewer #3: **Yes: **Professor Andrew P Smith

---

## [Editor Report · Acceptance letter]

12 Sep 2022

PONE-D-21-21515R2 

Wellbeing and Coping of UK Nurses, Midwives and Allied Health Professionals during COVID-19-a cross sectional study 

Dear Dr. Gillen:

I'm pleased to inform you that your manuscript has been deemed suitable for publication in PLOS ONE. Congratulations! Your manuscript is now with our production department. 

Kind regards, 

on behalf of

Dr. Rosemary Frey 

Academic Editor

PLOS ONE